# COVID-19 Lockdown: Impact on Oral Health-Related Behaviors and Practices of Portuguese and Spanish Children

**DOI:** 10.3390/ijerph192316004

**Published:** 2022-11-30

**Authors:** Ana L. Costa, Joana L. Pereira, Lara Franco, Francisco Guinot

**Affiliations:** 1Paediatric and Preventive Dentistry Institute, Faculty of Medicine, University of Coimbra, 3000-075 Coimbra, Portugal; 2Center for Innovation and Research in Oral Sciences, Faculty of Medicine, University of Coimbra, 3000-075 Coimbra, Portugal; 3Department of Pediatric Dentistry, Faculty of Dentistry, Universitat Internacional de Catalunya, 080195 Sant Cugat del Vallès, Spain

**Keywords:** COVID-19, pediatric dentistry, oral health, food habits, behavior, family functioning

## Abstract

This study aimed to assess and compare the impact of COVID-19 pandemic lockdowns on the oral health attitudes, dietary habits and access to dental care of Portuguese and Spanish children. A cross-sectional observational study involving caregivers of 3–17-year-old children who had cohabited during a COVID-19 pandemic lockdown in Spain and Portugal was conducted. Caregivers completed an online anonymous questionnaire. Aiming groups comparison, chi-square test was used for qualitative variables. 770 surveys were obtained. Significant changes in the children’s routine were higher in Portugal (*p* < 0.001). Both countries showed a large percentage of children who had between 2–3 snacks between meals (*p* < 0.001) and a higher consumption of snacks was particularly noticed among Spanish children with untreated dental caries during the lockdown (*p* = 0.003). Most caregivers reported children’s oral hygiene habits did not suffer noteworthy alterations (*p* = 0.417), although parental supervision of toothbrushing was associated with dental attendance during the lockdown. The majority of the sample had no dental attendance during confinement. Confinement appears to have not markedly affected the oral health status and habits of the majority of these children, although an important impact of some demographic and behavioral factors upon dietary and oral care/habits was detected.

## 1. Introduction

Coronavirus Disease 2019 (COVID-19), an infectious disease caused by the severe acute respiratory syndrome coronavirus 2 (SARS-CoV-2), was declared a global pandemic by the World Health Organization (WHO) in March 2020 [1]. The virus presents a remarkable infectivity potential, since it spreads through the inhalation of infected particles in droplets and aerosols and through direct contact with oral, ocular and nasal mucous membranes [2,3,4,5].

Aiming to contain the COVID-19 outbreak several countries declared a state of emergency and implemented measures involving social distancing, including quarantines and nationwide lockdown periods [5]. Social distancing, interruption of face-to-face education activities and home confinement have produced considerable lifestyle changes among children and their families worldwide [6].

In addition, since dental settings could be favorable for COVID-19 transmission due to several specificities of dental care procedures, significant restrictions were imposed on oral health care, as many nations instructed dental professionals to postpone elective treatments, to provide only urgent and emergency procedures and to adopt a variety of protocols to reduce the transmission risk [7,8,9,10,11,12]. Furthermore, fear of contracting COVID-19 appears to have produced an additional negative impact on the demand of care during confinement periods, even in emergency circumstances [3,5,6,13,14,15]. Suspension or deferral of routine diagnostic, preventive and curative care may have, therefore, endorsed an increased burden of oral disease across nations [16], thus emphasizing the need for maintaining oral preventive care in the home environment [17].

Overall, although it is conceivable that a profound impact on children’s oral condition and oral health-related behaviors, in particular oral hygiene and food-related habits, may have occurred, scarce studies have addressed these issues to date [2,5,6,18,19,20,21]. A limited number of countries, namely Brazil [2,6,22], China [23], Australia [16], Israel [24], Poland [25], Nigeria [3], have reported a decrease in the number of pediatric patients subjected to dental treatment during COVID-19 confinement. Difficulties accessing dental care [17] and a higher impact on dental attendance among vulnerable children from lower socioeconomic backgrounds [16,22] have also been described among some populations.

Reported eating habits have experienced fairly conflicting variations during lockdown, although a tendency towards overeating and an increased consumption of high-sugar processed food has been associated with sedentarism and to negative psychological states, expectably experienced by children and their families during these periods [2,6,18,20,26,27,28,29,30].

The limited evidence regarding oral hygiene practices during confinements is also contrasting, with few reports stating hygiene habits were maintained [6,17] and others revealing overall poor oral hygiene attitudes and practices among children [2,5]. A study performed in an early stage of the pandemics concluded that children who exhibited sleep disorders during social distancing tended to present poorer oral hygiene levels [31]. Nonetheless, one study conducted in Italy found a positive association between the increased presence of parents at home and the time spent on oral hygiene [32].

Assuming that caregivers exert a decisive influence in terms of health education and in ensuring preventive measures in their children [5,18], this study aims to retrospectively assess and compare, for the first time to our knowledge, the impact of the COVID-19 confinement on the oral health attitudes, access to dental care and dietary habits in Portuguese and Spanish children, as well as the caregiver’s perception of children’s oral health condition during this period.

## 2. Materials and Methods

The present cross-sectional observational study involved caregivers of 3–17-year-old children who had cohabited during a lockdown period of the COVID-19 pandemic in Spain and Portugal. Independent ethics committees from two universities, one public Portuguese (Faculty of Medicine, University of Coimbra, Portugal, project reference CE-076/2021; 16 June 2021) and other private Spanish institution (CER, International University of Catalonia, Barcelona, Spain, project reference ODP-ECL-2021-04; 14 June 2021) approved the study protocol, which complied with the Declaration of Helsinki (version 2002, 2013 revision) for human research. Caregivers who consented to participate completed an anonymous questionnaire, available online from June to December 2021. The anonymous nature of this questionnaire did not allow tracing sensitive personal data in any way.

The semi-structured survey (Appendix A), based on previous studies [2,6,26,31], was written in Spanish and Portuguese and developed on Office 365 and Google Forms platforms. The questionnaire, completed within approximately 5 min, was divided into 5 main domains to retrospectively assess: (1) participants’ sociodemographic characteristics (12 questions); and child’s (2) routine changes (3 questions); (3) dietary habits (4 questions); (4) oral hygiene practices (7 questions); (5) access to dental care and the caregiver’s perception of the child’s oral health status (10 questions), during the last COVID-19 lockdown period experienced in each country. Participants who had not been the children’s main caregivers during their last confinement period or who did not fluently understand Spanish or Portuguese were excluded.

A pilot study was performed on a small randomly selected sample (n = 30) to verify the functionality of the questionnaire. These participants were not included in the final sample.

The study was conducted using a snowball sampling technique, as the survey was disseminated through the researchers’ personal and professional networks, along with schools that agreed to forward the recruitment link to parents.

Sample size determination was calculated considering a 95% confidence interval and a 5% precision. About 380 participants were settled as minimum sample size for each country.

Data was exported and analyzed using the Statistical Package for the Social Sciences (SPSS, version R 4.1.2, SPSS Inc., Chicago, IL, USA) using adequate statistical tests at a 95% confidence level. Qualitative variables were described by frequency and percentage. For the comparison of two groups, the chi-square test was used for qualitative variables. The tests were considered significant for a *p*-value less than 0.05.

This study has been designed to meet STROBE guidelines.

## 3. Results

### 3.1. Demographic Data

A total sample of 770 surveys was obtained, 384 in Spain and 386 in Portugal. The study sample presented a similar distribution regarding the gender of children involved in the surveys (*p* = 0.221). The age of the children ranged from 3 to 17 years, with the group from 3 to 7 years being the most prevalent in Spain (44.5%) and from 8 to 12 years in Portugal (43.5%). Regarding the place of residence, in Spain 88% of the respondents lived in an urban area while in Portugal the percentage was 52.8% (*p* < 0.001). Concerning the educational level of the child’s main caregiver, the percentage with the highest educational level was 28.8% in Spain and 21.3% in Portugal. On the other hand, the percentage with a lower educational level was 1.3% and 0.3%, respectively, finding a statistically significant difference (*p* = 0.002) between both nationalities. Study demographics are shown in Table 1.

### 3.2. Impact of Confinement on the Lifestyle of the Child and the Family

Regarding the employment status of the main caregiver during confinement, statistically significant results were observed (*p* < 0.001). In Spain 30.2% were working full time from home, while in Portugal 37.3%. On the other hand, in Spain 13.3% were working part-time from home, while in Portugal the percentage was 0%. Regarding caregivers who were not working, both countries presented a similar distribution in the study (21.4% vs. 21.8%). During confinement, 76.8% of Spanish children lived with siblings at home, while in Portugal 69.4% (*p* = 0.021). With respect to family income, in both countries, over half of the respondents reported that they did not suffer any impact, however, 15.2% in Spain and 7.4% in Portugal revealed that they suffered a drastic reduction, this data set being statistically significant (*p* = 0.012). The percentage of respondents who noticed significant changes in the children’s life routine during confinement was higher in Portugal than in Spain (78.2% vs. 58.9%), with a statistically significant difference (*p* < 0.001). The data on the impact of confinement on the lifestyle of the child and the family in the study are shown in Table 1. About 32.1%, 27.0% and 18.3% of the caregivers from both nations admitted their children experienced emotional states of annoyance, nervousness and disinterest during confinement periods.

### 3.3. Frequency of Food Consumption during Confinement

Both countries showed a large percentage of children who had 2 to 3 snacks in between meals, with a statistically significant difference between the two countries (*p* < 0.001). With regard to consumption of chocolate and candies 2 to 3 times a day, Portugal presented a higher percentage (17.1%) compared to Spain (13.5%) (*p* < 0.001). Regarding the consumption of sweets and bakery products, a higher frequency of daily consumption was observed in Portugal (*p* = 0.119). In Spanish children, an excessive consumption of candy and chocolate was observed in those whose parents suffered income losses during confinement (*p* = 0.103). On the contrary, an excessive consumption of these products was noticed in Portuguese families that did not suffer income losses (*p* = 0.551).

As to consumption of cereals, a lower frequency was observed in Spain (*p* = 0.002). Within the daily consumption of juices, homemade juices were the most consumed by children in both countries (*p* = 0.046). In this matter, it was possible to verify an excessive and significant (*p* = 0.009) consumption of industrial fruit juice in Spanish children whose families had suffered income losses during the lockdown period. With regard to Portuguese children, an excessive consumption of these beverages was noticed among families who did not experience income losses, albeit not significantly.

63.5% of the Spanish respondents consumed dairy products 2–3 times a day, while in Portugal, the percentage was 53.4% (*p* = 0.002). An overview of the reported eating habits during the confinement is shown in Table 2 (and Table 4).

### 3.4. Behaviors and Oral Health Status during Confinement

74.3% of Spanish and 72.2% of Portuguese caregivers reported that their perception was that their children’s oral hygiene habits had not suffered significant alterations (*p* = 0.417). Nonetheless, it was found in both countries that families of children with worse oral hygiene suffered income losses during confinement, without statistically significant difference in both cases.

Respecting the frequency of toothbrushing, 46.4% of Spanish participants and 58.6% of Portuguese participants stated that their children brushed twice a day (*p* = 0.002). However, the survey revealed that only 12.9% of Spanish children and 14.3% of Portuguese children increased the frequency of toothbrushing during confinement, comparing to the previous period (*p* = 0.855). In addition, only 8.6% in Spain and 18.5% in Portugal indicated that they used dental floss once a day, showing a statistically significant difference (*p* < 0.001). In Spain, 35.7% of caregivers and 35.8% in Portugal stated that they did not control their children’s toothbrushing nor were present at the time of brushing (*p* = 0.852). It was found that the higher percentage of Spanish children with poorer oral hygiene belonged to families in which the main’s caregiver presented a lower level of education, contrary to what was verified among Portuguese participants, but the differences were not statistically significant in either of the cases.

Although most of both Spanish and Portuguese children had not untreated oral conditions during lockdown, a non-negligible number of children presented untreated dental caries lesions or gingival inflammation/bleeding. The percentage of children who presented conditions such as dental trauma, halitosis and toothache during confinement due to not having been treated previously was markedly low.

In both Spain and Portugal, children who had untreated dental caries during the lockdown period had worse oral hygiene practices in the same period and these differences were statistically significant among Portuguese participants (*p* = 0.002). Furthermore, it was possible to verify that both Portuguese and Spanish children who had untreated dental caries consumed a greater number of daily snacks in between meals, data being statistically significant in Spain (*p* = 0.003).

The majority of parents (71.4%) in both countries stated that their children did not visit the dentist during confinement. Only 35.3% of Portuguese children visited the dentist, while in Spain this percentage dropped to 21.9% (*p* < 0.001). Of those children who visited the dentist, most did so for a routine visit (*p* < 0.001) or for a previously scheduled treatment (*p* = 0.033). Likewise, the percentage of children who declared not going to the dentist for fear of transmitting the virus was similar in both countries (*p* = 0.850). However, the percentage of children who did not attend because the clinics had suspended their activity was 9.4% in Spain, increasing to 14% in Portugal (*p* = 0.046). Either in Spain or Portugal, children who reportedly required dental care during confinement did not decrease the frequency of visits, being these data statistically significant for Spanish children (*p* = 0.003).

In Spain, children who required dental care during lockdown period exhibited a greater ingestion of candy or chocolate in that time interval (*p* = 0.389); nonetheless this finding was not observed with regard to Portuguese children. Children from both countries who needed dental care during confinement were those with excessive ingestion of industrial fruit juice, these being statistically significant data in Spain (*p* < 0.001). In relation to number of daily snacks in between meals, it doesn´t seem to exert a significative difference in what concerns dental attendance. However, in children of both countries, it was observed that dental care during confinement was significantly related with parental supervision during toothbrushing.

Study data on dental hygiene, untreated oral conditions and dental care behaviors are shown in Table 3 and Table 4.

## 4. Discussion

To our best knowledge, this was the first study that concertedly assessed and compared oral health-related behaviors and practices of Portuguese and Spanish 3–17-year-old children during COVID-19 lockdown. Interestingly, and being geographically neighboring countries, lifestyles and behaviors are not usually transposable showing, however, very similar results in most of the items evaluated in this study. To date, as far as we know, only one study has studied parental perception of children’s oral hygiene during lockdown in Portugal [31]. On the other hand, some reports from Spain have explored the influence of confinement on diet and other lifestyle behaviors, though not focusing specifically on oral health matters [27,30]. In the current study, a wide variety of work status was observed among caregivers during the lockdown and most of the children cohabited with siblings throughout this period. Over half of those surveyed from both countries stated the family income was not impacted during the confinement, whilst a third of the participants mentioned the total income was slightly reduced. Although the majority of caregivers from both countries felt children experienced significant changes in their lifestyle routine during the lockdown, this proportion was considerably higher among respondents from Portugal (*p* < 0.001). Overall, these findings are consistent with earlier observations from other countries, which have evaluated the effects of the confinement on lifestyle behaviors such as sleep time, oral physical activity, dietary patterns, oral hygiene practices and healthcare-related attitudes [2,6,17,20,27,31,32,33,34,35,36]. Previous studies from other countries have found conflicting evidence on changes in dietary habits during the lockdown: whilst some reported an improvement on eating patterns [37,38], others observed an increase in food intake and higher frequency of consumption of ultra-processed foods with low nutritional value and high sugar content [6,17,26,27,28,34]. Negative impact on dietary habits has been attributed to several factors, including remote school for children, working from home for parents, reduced income and negative emotions as stress or anxiety [6,20,26,34]. In the current study, a surprising finding was that, in both countries, a reduced number of snacks were ingested in between meals during the lockdown, with most children eating only two snacks per day. Only a minority of the total sample (7.1%) ingested 5 or more snacks a day. Research has shown that the type and frequency of fermentable carbohydrates’ intake exert a critical impact on caries onset and progression [39,40].

Most respondents from both countries reported limiting consumption of sugary snacks as candy/chocolates and cake/sweet cookies to 1 to 3 times a week during confinement, followed by only once a week. Interestingly, while results relative to sugary cereals/cereal bars were also similar in both countries, as most indicated their children consumed such foods on a weekly basis, almost 20% of the sample ingested it 2 to 3 times a day. This might be considered a worrying finding, considering such snacks not merely present high sugar content but also are potentially adhesive to dental surfaces, enabling prolonged contact between teeth and fermentable substrate. Yet, in terms of dietary patterns, an excessive ingestion of industrial fruit juice was observed in both Portugal and Spain in children whose parents had lower education levels. Moreover, it should be mentioned that overall data appears to indicate that children belonging to families with income losses during the lockdown tended to excessively consume sweets, chocolates and industrial juices (in Spain). Constraints and difficulties accessing fresh products daily during the lockdown may explain the reduction in consumption of such products (including fruit, for example), at the expense of highly processed products, rich in sugar, salt and with a low level of nutrients [28,34,41]. At the same time, it seems plausible to speculate that the appreciation of healthy food may also have been relegated to the background, given the emotional instability and uncertainties in this unique period of families’ lives, as well as the conditioned time of caregivers.

Overall, the perception of most caregivers, whether Portuguese or Spanish, was that children’s oral hygiene practices were not significantly altered during the lockdown, in comparison to the pre-pandemic period, followed by a proportion of 15.2% stating oral hygiene practices had worsened during the lockdown, results which are consistent with data obtained in earlier studies [17,31]. The majority of participants from both countries claimed children maintained toothbrushing frequency during the lockdown. It should be noted that about 5% of the children in the sample did not brush their teeth daily, although the majority (52.5% of total respondents) complied with it, as reported, twice a day, though a non-neglectful part of all (35.7%) without parental supervision or assistance. The vast majority of children included in this sample had no regular flossing habits.

Once again, the family income during the lockdown and the main caregiver’s level of education should be assumed as important factors in this matter, taking in account data overall indicate that children of both countries belonging to families with losses of income during this time interval tended to worsen their oral hygiene habits. Among Spanish participants, the highest percentage of children with poorer oral hygiene belonged to families with lower education levels; however, in Portugal the opposite was observed: children with better oral hygiene had caregivers with higher levels of education, although these data were not statistically significant.

Our data also suggested that children who had untreated dental caries during the lockdown period had worse hygiene practices (Portugal, *p* = 0.002) and consumed a greater quantity of daily snacks (Spain, *p* = 0.003). It would be plausible that changes in family routines, including the need to adapt to work responsibilities, concomitantly with an increase in domestic activities, caring for children and anxiety related to a new infection spreading worldwide and being confined at home could have interfered with children’s sleep and oral hygiene habits, as previously reported [31,33,42]. The particular case of younger children, who rely on parental help to perform oral health measures, changes in routines could have had a more significant impact [31]. Nonetheless, it is possible that a subjective parameter such as parental perception of children´s oral hygiene may have led to different interpretations by caregivers: some respondents may have assumed the question was relative to the time spent toothbrushing, rather than its quality or frequency.

General limitations in terms of accessibility and even imposition of the type of medical and dental treatments that, at an early stage, at the beginning of the pandemic, in several countries including Portugal and Spain, included only response to clinical conditions considered urgent, as well as the fear of being infected during oral treatment have been pointed out in the literature as some of the reasons behind the reduction of care provided during the COVID-19 pandemic. It is considered that the neglect, even transitory, of any kind of dental treatment, including preventive care and routine appointments may contribute to an increase in the incidence and/or worsening of oral pathologies generically across the population, but eventually more pronouncedly in groups considered to be more susceptible. Some of these conclusions derive from the results reported in different countries during and even after confinement [3,6,16,23,24,43,44]. The role of different cellular effectors (e.g., mast cells) in the inflammatory response in the oral cavity, including the integrity of periodontal tissues in which oral hygiene is determinant in children and adults, should be also highlighted given the potential that SARS-CoV-2 infection may represent at this level, as discussed in the literature [45].

Brondani et al. stated that there was a significant decrease in the frequency of toothbrushing, dental attendance and self-perception of dental treatment need during pandemic period [2,17] also added the concept of influence of “family functioning” (better family functioning during lockdown was associated with improved family hygiene and nutrition, parental resilience and lower mental stress among children, as reported by their parents) [17].

In this study the majority of participants from both nations referred that there was no change in the frequency of access to pediatric dentistry consultations during lockdown and that neither fear of disease transmission (for about 95% of Portuguese and Spanish respondents) nor the suspension of clinical care activity were reasons for non-assistance (*p* = 0.046). However, our results were in line with some earlier findings, as about 71.4% of respondents of both countries mentioned having had no attendance during confinement, with statistically significant differences, when the question referred specifically to routine visits (*p* < 0.001) and to pre-scheduled treatments (*p* = 0.033). Spanish children who needed dental care during confinement tended to excessively consume candy/chocolates during the lockdown, without these data being statistically significant, a finding not verified in the Portuguese sample. In this same item, there was a tendency among Spanish children to eat a higher number of snacks in between meals and towards a higher consumption of industrial fruit juice (*p* < 0.001). Among both Spanish and Portuguese participants, the ones who needed more dental care during this period were those without parental supervision in toothbrushing (*p* < 0.005). These last results also seem to be broadly in line with the findings described in the literature, as mentioned above, with the occasional differences, more or less marked, between the two countries most likely related to the financial conjecture and somewhat different social/family dynamics.

One major limitation of this cross-sectional study naturally concerns the use of a snowball sampling technique, which may possibly have contributed to selection bias, hampering the generalization of our findings to the pediatric population of both countries. It is plausible that, for instance, our overall favorable findings regarding dietary choices, arising mainly from children living in urban areas whose most caregivers had attended academic education, possibly indicating higher socioeconomic position, may not be straightforwardly representative of the habits of children from Spain and Portugal, particularly since earlier reports from Spain and Brazil have showed lower class isolated families presented lower consumption of healthy foods during the period of confinement [29,30].

Nevertheless, this sampling strategy allowed conducting the study in both countries within important time restraints: the risk of recall bias would have increased significantly in case this retrospective survey had not been conducted within a short period of time after the last COVID-19 confinements, potentially leading to inaccurate data collection.

Another potential source of bias may be related to social desirability bias, since it is known that in questionnaire-based studies some respondents may have a tendency to answer in a more socially acceptable manner. Moreover, the fact that this questionnaire was made available for completion online, remotely, with a limited number of questions, without contact between the respondent and the researchers, and that caregivers filled in the questionnaire reporting their responses to data specifically related to their dependents (and not themselves) list factors that may contribute to some degree of bias in the answers and consequent analysis. Still, the design of this online questionnaire followed the structure of other surveys recently conducted on this matter. Moreover, the simultaneous collection of comparative data pertaining to different time-points (confinement vs. non confinement) might have led to inaccurate answers and, due to social distancing measures, clinical assessment of the children’s oral hygiene and oral health status was not possible [30,31]. Accordingly, the present study did not intend to perform a clinical evaluation of the participating children, but to assess the caregivers’ perception on the oral health needs and status of their children.

## 5. Conclusions

The present study has provided, for the first-time, as far as we know, comparative data on the eating patterns, oral health-related behaviors and access of dental care of Portuguese and Spanish pediatric populations during the COVID-19 lockdown. Within the limitations of this study, present findings suggest confinement due to the COVID-19 pandemic did not markedly affect the oral health status and habits of the majority of children from both nations, according to the perception of their caregivers. However, an overall marked effect of demographic factors as loss of family income and level of education of caregivers was noticed regarding dietary and oral hygiene habits.

The present results also indicate that a wide majority of the surveyed children was not subjected to professional dental care, although children who needed dental care the most during those periods did not have parental supervision while toothbrushing. Accordingly, it is thus possible that some harmful effects of the pandemic on oral health among these populations may be yet to be fully portrayed. As a consequence, and considering the COVID-19 pandemic is still ongoing, some of our data need to be additionally explored in future larger population studies.

## Figures and Tables

**Table 1 ijerph-19-16004-t001:** Study sample characterization: sociodemographic features and impact of the lockdown on the child and family’s lifestyle.

Variable	COUNTRY		Total Sample n (%)	*p*-Value
	Spain n (%)	Portugal n (%)		
**Gender**				
Female	187 (48.7%)	205 (53.1%)	392 (50.9%)	0.221
Male	197 (51.3%)	181 (46.9%)	378 (49.1%)	
**Total**	384 (100%)	386 (100%)	770 (100%)	
**Age (years)**				
3–7	171 (44.5%)	132 (34.2%)	303 (39.4%)	**0.006**
8–12	153 (39.8%)	168 (43.5%)	321 (41.7%)	
13–17	60 (15.6%)	86 (22.3%)	146 (19.0%)	
**Place of residence**				
Rural	46 (12%)	182 (47.2%)	228 (29.6%)	**<0.001**
Urban	338 (88%)	204 (52.8%)	542 (70.4%)	
**Mother’s educational level**				
Group I ^1^	48 (13.2%)	41 (11.8%)	89 (12.5%)	**0.036**
Group II ^2^	149 (40.8%)	139 (40.2%)	288 (40.5%)	
Group III ^3^	67 (18.4%)	43 (12.4%)	110 (15.5%)	
Group IV ^4^	64 (17.5%)	65 (18.8%)	129 (18.1%)	
Group V ^5^	37 (10.1%)	58 (16.8%)	95 (13.4%)	
**Father’s educational level**				
Group I	59 (16.5%)	45 (13.3%)	104 (14.9%)	**<0.001**
Group II	148 (41.3%)	101 (29.9%)	249 (35.8%)	
Group III	71 (19.8%)	59 (17.5%)	130 (18.7%)	
Group IV	49 (13.7%)	70 (20.7%)	119 (17.1%)	
Group V	31 (8.7%)	63 (18.6%)	94 (13.5%)	
**Main caregiver’s level of education**				
Master’s, postgraduate or doctoral degree	109 (28.8%)	81 (21.3%)	190 (25.0%)	**0.002**
Bachelor degree	163 (43.1%)	155 (40.7%)	318 (41.9%)	
Basic or secondary education	101 (26.7%)	144 (37.8%)	245 (32.3%)	
Illiterate	5 (1.3%)	1 (0.3%)	6 (0.8%)	
**Main caregiver’s work status during the lockdown**				
At home, not working	82 (21.4%)	84 (21.8%)	166 (21.6%)	**<0.001**
Working full-time from home	116 (30.2%)	144 (37.3%)	260 (33.8%)	
Working part-time from home	51 (13.3%)	0 (0.0%)	51 (6.6%)	
Working full-time outside the house	67 (17.4%)	91 (23.6%)	158 (20.5%)	
Working part-time outside the house	51 (13.3%)	53 (13.7%)	104 (13.5%)	
Other	17 (4.4%)	14 (3.6%)	31 (4.0%)	
**Siblings in household during the lockdown**				
Yes	295 (76.8%)	268 (69.4%)	563 (73.1%)	**0.021**
No	89 (23.2%)	118 (30.6%)	207 (26.9%)	
**Family income during the lockdown**				
Increased	9 (2.4%)	7 (1.8%)	16 (2.1%)	**0.012**
Not impacted	193 (51.3%)	206 (54.4%)	399 (52.8%)	
Slightly reduced	115 (30.6%)	137 (36.1%)	252 (33.4%)	
Drastically reduced	57 (15.2%)	28 (7.4%)	85 (11.3%)	
Total loss	2 (0.5%)	1 (0.3%)	3 (0.4%)	
**Significant changes in child’s lifestyle routine during the lockdown**				
Yes	226 (58.9%)	302 (78.2%)	528 (68.6%)	**<0.001**
No	158 (41.1%)	84 (21.8%)	242 (31.4%)	

Statistically significant (*p* < 0.05) differences in bold. ^1^ Senior managers/owners in large public/private business organizations, university lecturers, defense forces commissioned officers, qualified professionals or occupations in government administration. ^2^ Other medium-sized business owners/managers, associate professionals, defense forces officers or basic/secondary education teachers. ^3^ Tradesmen/women, clerks and skilled office, sergeants (armed forces) or related occupations. ^4^ Small farmer, worker in agriculture or in related fields, office assistants, machine operators, and defense forces without qualification not included above. ^5^ Unpaid workers, laborer, related workers without qualifications or occupations not classified above.

**Table 2 ijerph-19-16004-t002:** Dietary habits during the lockdown: frequency of snacks in between meals and types of foods and drinks consumed during snacks.

Variable	COUNTRY		Total Sample n (%)	*p*-Value
	Spain n (%)	Portugal n (%)		
**Number of daily snacks in between meals**				
5 or more snacks a day	31 (8.5%)	21 (5.7%)	52 (7.1%)	**<0.001**
Four snacks	33 (9.1%)	56 (15.3%)	89 (12.2%)	
Three snacks	49 (13.5%)	106 (29.0%)	155 (21.3%)	
Two snacks	153 (42.1%)	147 (40.2%)	300 (41.2%)	
One snack	97 (26.7%)	36 (9.8%)	133 (18.2%)	
**Candy, chocolate**				
1–3 times a week	193 (50.3%)	192 (49.7%)	385 (50.0%)	**<0.001**
2–3 times a day	53 (13.8%)	71 (18.4%)	124 (16.1%)	
Four or more times a day	0 (0.0%)	3 (0.8%)	3 (0.4%)	
Less than once a week	138 (35.9%)	120 (31.1%)	258 (33.5%)	
**Cake, sweet cookies, ice cream**				
1–3 times a week	199 (51.8%)	214 (55.4%)	413 (53.6%)	0.119
2–3 times a day	52 (13.5%)	66 (17.1%)	118 (15.3%)	
Four or more times a day	1 (0.3%)	2 (0.5%)	3 (0.4%)	
Less than once a week	132 (34.4%)	104 (26.9%)	236 (30.6%)	
**Sugary cereals, cereal bars**				
1–3 times a week	115 (29.9%)	153 (39.6%)	268 (34.8%)	**0.002**
2–3 times a day	66 (17.2%)	78 (20.2%)	144 (18.7%)	
Four or more times a day	2 (0.5%)	0 (0.0%)	2 (0.3%)	
Less than once a week	201 (52.3%)	155 (40.2%)	356 (46.2%)	
**Homemade fruit juice**				
1–3 times a week	108 (28.1%)	123 (31.9%)	231 (30.0%)	**0.046**
2–3 times a day	52 (13.5%)	30 (7.8%)	82 (10.6%)	
Four or more times a day	0 (0.0%)	1 (0.3%)	1 (0.1%)	
Less than once a week	224 (58,3%)	232 (60.1%)	456 (59.2%)	
**Industrial fruit juice**				
1–3 times a week	67 (17.4%)	82 (21.2%)	149 (19.4%)	0.151
2–3 times a day	45 (11.7%)	29 (7.5%)	74 (9.6%)	
Four or more times a day	1 (0.3%)	2 (0.5%)	3 (0.4%)	
Less than once a week	271 (70.6%)	273 (70.7%)	544 (70.6%)	
**Soda**				
1–3 times a week	63 (16.4%)	39 (10.1%)	102 (13.2%)	**0.003**
2–3 times a day	14 (3.6%)	26 (6.7%)	40 (5.2%)	
Four or more times a day	4 (1.0%)	0 (0.0%)	4 (0.5%)	
Less than once a week	303 (78.9%)	321 (83.2%)	624 (81.0%)	
**Plain milk, dairy**				
1–3 times a week	102 (26.6%)	107 (27.7%)	209 (27.1%)	**0.002**
2–3 times a day	244 (63.5%)	206 (53.4%)	450 (58.4%)	
Four or more times a day	3 (0.8%)	10 (2.6%)	13 (1.7%)	
Less than once a week	35 (9.1%)	63 (16.3%)	98 (12.7%)	

Statistically significant (*p* < 0.05) differences in bold.

**Table 3 ijerph-19-16004-t003:** Oral health behaviors and status during the lockdown: dental hygiene, untreated oral conditions and dental attendance patterns.

Variable	COUNTRY		Total Sample n (%)	*p*-Value
	Spain n (%)	Portugal n (%)		
**Parental perception on child’s oral hygiene habits**				
Unchanged	274 (74.3%)	270 (72.2%)	544 (73.2%)	0.417
Improved	37 (10.0%)	49 (13.1%)	86 (11.6%)	
Worsened	58 (15.7%)	55 (14.7%)	113 (15.2%)	
**Toothbrushing frequency**				
Not daily	27 (7.0%)	14 (3.6%)	41 (5.3%)	**0.002**
Once a day	110 (28.6%)	80 (20.8%)	190 (24.7%)	
Twice a day	178 (46.4%)	225 (58.6%)	403 (52.5%)	
More than twice a day	69 (18.0%)	65 (16.9%)	134 (17.4%)	
**Changes in toothbrushing frequency**				0.855
Unchanged	270 (71.2%)	264 (70.0%)	534 (70.6%)	
Increased	49 (12.9%)	54 (14.3%)	103 (13.6%)	
Decreased	60 (15.8%)	59 (15.6%)	119 (15.7%)	
**Parental supervision while toothbrushing**				
Daily	113 (29.9%)	121 (31.6%)	234 (30.7%)	0.852
Frequently	56 (14.8%)	59 (15.4%)	115 (15.1%)	
Occasionally	74 (19.6%)	66 (17.2%)	140 (18.4%)	
Never	135 (35.7%)	137 (35.8%)	272 (35.7%)	
**Flossing frequency**				
Not daily	328 (90.6%)	274 (78.1%)	602 (84.4%)	**<0.001**
Once a day	31 (8.6%)	65 (18.5%)	96 (13.5%)	
Twice a day	2 (0.6%)	11 (3.1%)	13 (1.8%)	
More than twice a day	1 (0.3%)	1 (0.3%)	2 (0.3%)	
**Untreated oral conditions**				
**Dental caries**				
No	309 (80.5%)	343 (88.9%)	652 (84.7%)	**0.001**
Yes	75 (19.5%)	43 (11.1%)	118 (15.3%)	
**Gingival swelling or bleeding**				
No	291 (75.8%)	294 (76.2%)	585 (76.0%)	0.901
Yes	93 (24.2%)	92 (23.8%)	185 (24.0%)	
**Dental trauma**				
No	383 (99.7%)	384 (99.5%)	767 (99.6%)	0.566
Yes	1 (0.3%)	2 (0.5%)	3 (0.4%)	
**Halitosis**				
No	375 (97.7%)	365 (94.6%)	740 (96.1%)	**0.026**
Yes	9 (2.3%)	21 (5.4%)	30 (3.9%)	
**Toothache**				
No	364 (94.8%)	354 (92.7%)	718 (93.7%)	0.226
Yes	20 (5.2%)	28 (7.3%)	48 (6.3%)	
**Dental attendance during the lockdown**				
No	300 (78.1%)	247 (64.7%)	547 (71.4%)	**<0.001**
Yes	84 (21.9%)	135 (35.3%)	219 (28.6%)	
**Attendance for routine visit**				
No	350 (91.1%)	307 (80.4%)	657 (85.8%)	**<0.001**
Yes	34 (8.9%)	75 (19.4%)	109 (14.2%)	
**Attendance for pre-scheduled treatment**				
No	352 (91.7%)	332 (86.9%)	684 (89.3%)	**0.033**
Yes	32 (8.3%)	50 (13.1%)	82 (10.7%)	
**Attendance for toothache**				
No	376 (97.9%)	376 (98.4%)	752 (98.2%)	0.596
Yes	8 (2.1%)	6 (1.6%)	14 (1.8%)	
**Attendance for dental trauma**				
No	383 (99.7%)	379 (99.2%)	762 (99.5%)	0.319
Yes	1 (0.3%)	3 (0.8%)	4 (0.5%)	
**Changes in dental attendance frequency during the lockdown**				
Unchanged	243 (64.6%)	272 (73.1%)	515 (68.9%)	**0.041**
Increased	11 (2.9%)	7 (1.9%)	18 (2.4%)	
Decreased	122 (32.4%)	93 (25.0%)	215 (28.7%)	
**No attendance due to fear of COVID-19 transmission**				
No	366 (95.3%)	369 (95.6%)	735 (95.5%)	0.850
Yes	18 (4.7%)	17 (4.4%)	35 (4.5%)	
**No attendance due to activity on clinics suspended**				
No	348 (90.6%)	332 (86.0%)	680 (88.3%)	**0.046**
Yes	36 (9.4%)	54 (14.0%)	90 (11.7%)	

Statistically significant (*p* < 0.05) differences in bold.

**Table 4 ijerph-19-16004-t004:** Sociodemographic characteristics, dietary patterns, oral health behaviors and status of the participating children during the lockdown.

COUNTRY	Variable		Variable		Total Sample n (%)	*p*-Value
			**Parental perception on child’s oral hygiene habits**			
			Worsened	Not worsened		
**Spain** n (%)	**Family income during the lockdown**	Not reduced	28 (14.1%)	170 (85.9%)	198 (100.0%)	0.284
		Reduced	30 (18.3%)	134 (81.7%)	164 (100.0%)	
**Portugal** n (%)		Not reduced	29 (14.1%)	176 (85.9%)	205 (100.0%)	0.612
		Reduced	26 (16.0%)	136 (84.0%)	162 (100.0%)	
**Spain** n (%)			**Ingestion of candy or chocolate during the lockdown**			
			Excessive	Moderate		
		Not reduced	22 (10.9%)	180 (89.1%)	202 (100.0%)	0.103
		Reduced	29 (16.7%)	145 (83.3%)	174 (100.0%)	
**Portugal** n (%)		Not reduced	41 (19.2%)	172 (80.8%)	213 (100.0%)	0.551
		Reduced	28 (16.9%)	138 (83.1%)	166 (100.0%)	
			**Ingestion of industrial fruit juice during the lockdown**			
			Excessive	Moderate		
**Spain** n (%)		Not reduced	16 (7.9%)	186 (92.1%)	202 (100.0%)	**0.009**
		Reduced	29 (16.7%)	145 (83.3%)	174 (100.0%)	
**Portugal** n (%)		Not reduced	18 (8.5%)	195 (91.5%)	213 (100.0%)	0.827
		Reduced	13 (7.8%)	153 (92.2%)	166 (100.0%)	
			**Parental perception on child’s oral hygiene habits**			
			Worsened	Not worsened		
**Spain** n (%)	**Main caregiver’s level of education**	High education	36 (13.7%)	226 (86.3%)	262 (100.0%)	0.098
		Low education	21 (20.8%)	80 (79.2%)	101 (100.0%)	
**Portugal** n (%)		High education	35 (15.2%)	196 (84.8%)	231 (100.0%)	0.696
		Low education	19 (13.7%)	120 (86.3%)	139 (100.0%)	
			**Untreated dental caries during the lockdown**			
			No	Yes		
**Spain** n (%)	**Parental perception on child’s oral hygiene habits**	Worsened	43 (74.1%)	15 (25.9%)	58 (100.0%)	0.145
		Not worsened	256 (82.3%)	55 (17.7%)	311 (100.0%)	
**Portugal** n (%)		Worsened	37 (67.3%)	18 (32.7%)	55 (100.0%)	**0.002**
		Not worsened	269 (84.3%)	50 (15.7%)	319 (100.0%)	
**Spain** n (%)	**Number of daily snacks in between meals**	Low-frequency	211 (84.4%)	39 (15.6%)	250 (100.0%)	**0.003**
		High-frequency	80 (70.8%)	33 (29.2%)	113 (100.0%)	
**Portugal** n (%)		Low-frequency	151 (82.5%)	32 (17.5%)	183 (100.0%)	0.293
		High-frequency	143 (78.1%)	40 (21.9%)	183 (100.0%)	
			**Dental attendance during the lockdown**			
			No	Yes		
**Spain** n (%)	**Changes in dental attendance frequency during the lockdown**	Decreased	106 (86.9%)	16 (20.0%)	122 (100.0%)	**0.003**
		Unchanged	186 (73.2%)	68 (26.8%)	254 (100.0%)	
**Portugal** n (%)		Decreased	67 (72.0%)	26 (28.0%)	93 (100.0%)	0.053
		Unchanged	168 (60.9%)	108 (39.1%)	276 (100.0%)	
**Spain** n (%)	**Ingestion of candy or chocolate during the lockdown**	Excessive	39 (73.6%)	14 (26.4%)	53 (100.0%)	0.389
		Moderate	261 (78.9%)	70 (21.1%)	331 (100.0%)	
**Portugal** n (%)		Excessive	49 (67.1%)	24 (32.9%)	73 (100.0%)	0.624
		Moderate	198 (64.1%)	111 (35.9%)	309 (100.0%)	
**Spain** n (%)	**Ingestion of industrial fruit juice during the lockdown**	Excessive	26 (56.5%)	20 (43.5%)	46 (100.0%)	**<0.001**
		Moderate	274 (81.1%)	64 (18.9%)	338 (100.0%)	
**Portugal** n (%)		Excessive	19 (63.3%)	11 (36.7%)	30 (100.0%)	0.874
		Moderate	228 (64.8%)	124 (35.2%)	352 (100.0%)	
**Spain** n (%)	**Number of daily snacks in between meals**	Low-frequency	200 (80.0%)	50 (20.0%)	250 (100.0%)	0.053
		High-frequency	80 (70.8%)	33 (29.2%)	113 (100.0%)	
**Portugal** n (%)		Low-frequency	119 (65.7%)	62 (34.3%)	181 (100.0%)	0.689
		High-frequency	116 (63.7%)	66 (36.3%)	182 (100.0%)	
**Spain** n (%)	**Parental supervision while toothbrushing**	Frequent	143 (84.6%)	26 (15.4%)	169 (100.0%)	**0.006**
		Infrequent	152 (72.7%)	57 (27.3%)	209 (100.0%)	
**Portugal** n (%)		Frequent	128 (71.9%)	50 (28.1%)	178 (100.0%)	**0.005**
		Infrequent	117 (58.2%)	84 (41.8%)	201 (100.0%)	

Statistically significant (*p* < 0.05) differences in bold.

## Data Availability

Additional data that support our findings are available from the corresponding author upon reasonable request.

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
