# Peer review of "COVID-19 Lockdown: Impact on Oral Health-Related Behaviors and Practices of Portuguese and Spanish Children"

_ijerph, 2022, doi:10.3390/ijerph192316004_

Round 1

Reviewer 1 Report

The authors need to provide substantial editorial revisions to the Results and Discussion sections of the manuscript. There are many run-on sentences, use of double negatives, and grammatical errors. For example, Lines 198-202,  216-219, and 301-305 need to be rewritten and clarified. Line 311 uses the term 'might could have'. In general, Lines 198-228 need to be edited for content understanding.

In the Conclusion section, the authors appropriately point out and address the possible study design shortfalls. 

Reviewer 2 Report

Introduction and discussion are correct

Materials and methods are well described and pertinent

Results are clearly described and very coherent with the materials and methods

CONCLUSION is correct and interesting.

The study should further emphasize the importance of hygiene in the oral health of adults and children, including the references indicated:

PubMed ID 34425666

PubMed ID 34425662

PubMed ID 34425659

Reviewer 3 Report

This is a well done cross-sectional observational study utilizing an online survey completed by caregivers of 3-17 year old children who cohabited during the COVID-19 lockdown period in Spain and Portugal. Potential biases in the snowball sampling technique were adequately addressed.

Comments:
1. In each table statistically significant differences (p<0.05) were supposed to be bolded. Tables 2 ,3, and 4 have some significant differences that were not bolded.

2. Page 5, line 172. "63/5% of the Spanish respondent consumed daity product 4 or more times a day while in Portugal, the percentage was 53.4% (p=002)." According to Table 2, those percentages are indicated for diary consumption 2-3 times a day. Please correct.

3. The title "COBID-19 lockdown: impact on children's attitudes, behaviors and oral health in Portugal and Spain" emphasizes children's attitudes which is not really addressed since the survey was completed by the caregiver's observation of children's behaviors. The discussion indicated the study compared oral health-related behaviors and practices of Portuguese and Spanish children, which may be more accurate. Consider a title revision.
